# Discussion and Demonstration of RF-MEMS Attenuators Design Concepts and Modules for Advanced Beamforming in the Beyond-5G and 6G Scenario—Part 1

**DOI:** 10.3390/s24072308

**Published:** 2024-04-05

**Authors:** Girolamo Tagliapietra, Flavio Giacomozzi, Massimiliano Michelini, Romolo Marcelli, Giovanni Maria Sardi, Jacopo Iannacci

**Affiliations:** 1Center for Sensors and Devices (SD), Fondazione Bruno Kessler (FBK), 38123 Trento, Italy; gtagliapietra@fbk.eu (G.T.); giaco@fbk.eu (F.G.); m.michelini-1@studenti.unitn.it (M.M.); 2CNR-IMM, 00133 Roma, Italy; romolo.marcelli@cnr.it (R.M.); giovannimaria.sardi@cnr.it (G.M.S.)

**Keywords:** RF-MEMS, attenuators, low actuation voltage, beamforming, 5G, B5G

## Abstract

This paper describes different variants of broadband and simple attenuator modules for beamforming applications, based on radio frequency micro electro-mechanical systems (RF-MEMS), framed within coplanar waveguide (CPW) structures. The modules proposed in the first part of this work differ in their actuation voltage, topology, and desired attenuation level. Fabricated samples of basic 1-bit attenuation modules, characterized by a moderate footprint of 690 × 1350 µm^2^ and aiming at attenuation levels of −2, −3, and −5 dB in the 24.25–27.5 GHz range, are presented in their variants featuring both low actuation voltages (5–9 V) as well as higher values (~45 V), the latter ones ensuring larger mechanical restoring force (and robustness against stiction). Beyond the fabrication non-idealities that affected the described samples, the substantial agreement between simulations and measurement outcomes proved that the proposed designs could provide precise attenuation levels up to 40 GHz, ranging up to nearly −3 dB and −5 dB for the series and shunt variants, respectively. Moreover, they could be effective building blocks for future wideband and reconfigurable RF-MEMS attenuators. In fact, in the second part of this work, combinations of the discussed cells and other configurations meant for larger attenuation levels are investigated.

## 1. Introduction

Variable attenuators represent an indispensable component in modern radio frequency (RF) devices or systems. Maintaining signal integrity while exerting control over signal power is a fundamental requirement for attenuators. Their predominant usage lies in applications demanding high linearity, low power consumption, and minimal temperature dependency, attributes that are almost unattainable with variable amplifiers [1]. Additionally, these components must meet criteria such as high dynamic range, low insertion loss, wide attenuation range, and broad frequency bandwidth [2]. 

Their applications span across automatic gain control amplifiers (AGC), impedance matching networks, and broadband vector modulators [3], and as a result, they are quite common in RF transceivers and front ends for full duplex wireless systems [4] and measurement devices [5]. In fact, attenuators are adopted in RF transceivers both at the transmitter and receiver side to correct variations in gain and signal strength [6], whereas they are meant to improve the isolation between ports in full duplex systems [7]. 

Concerning the current and future telecommunication scenario, variable attenuators play an important role in phased array antenna systems, a paradigm that characterizes the multiple-input-multiple-output (MIMO) antennas used in the base stations and in the small cells for the radio access of the current 5G network [8,9].

In such phased-array systems, ensuring gain control in each element is crucial for compensating gain and suppressing sidelobes [10]. A uniform array with identical gain in each element tends to exhibit a relatively high sidelobe level (SLL). Introducing gain tapering to the array elements results in reduced SLL, thereby reducing the impact of interfering signals, which is vital at least on the receiver side [11]. A broad tuning range is generally necessary when aiming for significant sidelobe suppression. For instance, the Taylor method mandates a tapering range of about 21 dB for a linear array when targeting an SLL below −30 dB [12,13]. Additionally, as the number of elements increases, the required gain/amplitude tuning resolution must also rise to minimize quantitative errors during gain tapering. Passive and active gain control topologies are available, namely attenuators and variable gain amplifiers (VGA), and the key parameters governing their design include the following: the gain tuning range and resolution, the insertion loss (for attenuators) or power gain (for VGAs), the DC power consumption, the associated phase error, and the input and output matchings.

On one hand, the use of all-passive components in attenuators results in zero DC power consumption [14], which is generally preferable, especially in the case of large antenna arrays. On the other hand, the allocation of spectrum for the current and future telecommunications scenario led to growing frequency carriers and to a growing amount of their bandwidth. This leads to an increasing miniaturization of both the radiating elements and raises the need for highly miniaturized and wideband components in their radio front ends [15].

Traditionally, RF variable attenuators were developed utilizing semiconductor device technologies, including RF complementary metal–oxide–semiconductor (CMOS) field-effect transistors [16], monolithic microwave integrated circuits (MMICs) [17], and PIN diodes [18]. Despite their effectiveness, these semiconductor-device-based variable attenuators, as active components, face inherent drawbacks such as elevated power consumption, limited power handling, and the generation of harmonics. These factors render them less favorable for applications demanding high linearity performance, especially at high frequencies [19]. Power saving in both the user terminal and base station, adopting active antennas for 5G and future mobile networks, is also a crucial objective. Devices based on materials like graphene and vanadium oxide have been considered in recent times. However, in the case of graphene, the observed insertion loss does not exhibit sufficiently flat levels across the considered frequency band [20], whereas, a high voltage is demanded to tune the attenuation of the structure based on vanadium oxide [21].

The RF micro electro-mechanical systems (RF-MEMS) are becoming progressively more popular in the development of the devices that are commonly found in modern radio front ends, since their low power consumption, insertion loss, together with their superior isolation and wideband linearity, make them a favorable candidate for microwave and millimeter-wave (mmWave) applications [22]. For this reason, different RF components were developed in the past years, ranging from switching matrixes [23], filters [24], and impedance matching networks [25], to phase shifters [26] and attenuators [27]. As compared to the other kind of devices, the field of RF-MEMS attenuators has been relatively less investigated to date. In fact, since the very first device presented in 2009 [28], less than two dozen devices of this kind have populated the scientific literature. Both digital and analog [2,3] implementations have been considered for the developed RF-MEMS attenuators, with the former representing the vast majority. Reconfigurability, broadband behavior, and good linearity represent the most sought-after features of RF MEMS attenuators. Within this context, research effort has been put into increasing the number of attenuation states, while maintaining a flat and stable impedance value across a broad spectrum of frequencies and a minimal error as compared to the desired attenuation level [29].

Generally, most of the existing RF-MEMS attenuators are based on a coplanar waveguide (CPW) transmission line and multiple metallic membranes, operated by electrostatic actuation and ohmic contact. They are meant to enable or disable the series or shunt resistors [30] (or both, combined in a Pi-shaped topology [27]) along the RF signal line, representing multiple and cascaded attenuation units. As an example, eight units comprising series and shunt resistors are placed alternately along the signal line in [31] to form a 256-states device. In some cases [28,32], such units are placed along different branches, selectable by single-pole-double-throw (SPDT) switches, while in others, sections consisting of two selectable RF lines (one attenuated and the other not attenuated) are cascaded [27,33,34].

The analog and digital devices developed so far find application in multiple frequency bands, with some ranging from DC up to 20 GHz [33,34], and others reaching 40 GHz [32] or 80 GHz [31]. Concerning the number of achievable attenuation states, most of the digital implementations consist in 3- or 4-bit architectures [27,33], while some reach 8 bits [31]. The maximum attenuation levels of the general-purpose devices developed so far are quite broad—most of the implementations attain almost −20 dB [2,28], while others achieve −35 dB [34] or even −70 dB [33]. Regarding the attenuation steps, they are quite small in case of analog devices (0.2 dB in [2]) and ampler in digital implementations (e.g., 5 dB in [34] and 10 dB in [33]), with small root-mean-square errors ranging from 0.2% [2] to 5% [34]. Despite such promising results, many of these general-purpose devices still present two limitations: on one hand, the pursuit of an ever-growing number of attenuation states led to devices with a substantial footprint or aspect ratio, while on the other hand, the use of electrostatic actuation often led to devices operated by a high bias voltage. In fact, if the 3.2 mm^2^ area reported in [34] is not alarming, footprints like 8.77 mm^2^, 2.15 × 7.5 mm^2^, or 8.28 × 2.37 mm^2^ of the devices, reported in [27,28,35], respectively, are not suitable for realistic mm-Wave MIMO systems. From a system level viewpoint, regardless of the brick or tiling architecture of the array [36], it may be problematic to fit such bulky devices in the current RF front ends or dense beamforming architectures serving the array of small radiating elements [37], each spaced by half-wavelength. Moreover, the integration of such attenuators with common CMOS-based electronic components (with driving voltages typically up to 6.5 V) could be problematic because of their substantially higher driving voltage, which would imply the use of step-up DC converters in the case of actuation voltages around 35 V [33], 45 V [31], or 51 V [35]. Such a need could be harmful especially in the case of large mm-Wave arrays, both in terms of space and costs. For this purpose, commercial variable attenuators based on RF-MEMS usually feature low driving voltages (3.3 V in [38]).

In this paper, we propose different variants of broadband and simple RF-MEMS attenuator modules featuring a reduced actuation voltage. The modules differ in their actuation voltage, topology, desired attenuation level, and number of states. From the lowest (in the 5–9 V range) to the highest driving voltage (45 V), different topologies of resistors (series and shunt) were optimized to attain different attenuation levels. Characterized by limited attenuation levels (−2, −3, −5 dB), the different modules were assessed as building blocks to develop more complex attenuators for beamforming applications, operating in the 24.25–27.5 GHz range, the spectrum allocated to 5G communications in Europe. This paper is organized as follows: After the brief description regarding the fabrication technology adopted at our facilities in Section 2, the electromechanical features of the membranes employed in the cells characterized by low actuation voltage are described in Section 3. In Section 4 basic attenuation cells featuring series and shunt resistors are described and critically assessed, both in their variants, with low and more marked actuation voltages. Finally, some definitive remarks conclude the first part of the present paper.

## 2. Fabrication

The manufacturing process at the FBK facility utilizes standard surface micromachining techniques on a high resistivity silicon substrate featuring eight masks/layers [39], as depicted in Figure 1. The substrate is a 6-inch silicon or quartz wafer, onto which a layer of silicon oxide is deposited. The initial layer is a 630 nm thick poly-silicon layer, deposited and patterned over areas designated for conductive paths or buried electrodes (Figure 1c). The resistivity of this layer varies depending on the boron dopant concentration, resulting in different possible resistivity values. The second layer consists of the subsequently deposited 300 nm silicon oxide layer, and in the openings facilitating electrical connections to the underlying poly-silicon layer. These openings will be filled with the same material used in the third layer: a 630 nm thick aluminum-based multi-metal layer. This multi-metal layer will be sputtered and patterned, and serve as either a buried conductive path for RF signals or an electrical connection with the first layer (Figure 1e).

The openings etched into the subsequently deposited 100 nm silicon oxide form the most important features of the fourth layer, necessary for establishing an electrical connection with the multi-metal. These openings will be filled with the material specified in the fifth layer. The fifth layer consists of 150 nm of gold, evaporated onto the openings to serve as an electrical link between the multi-metal and the electroplated gold layers composing the actual MEMS structure.

The sixth layer is characterized by a 3-μm photoresist sacrificial tier, patterned in areas where suspended MEMS structures will be located. The seventh mask involves 2-μm thick electroplated gold, positioned at RF signal lines, ground, various pads, and the suspended membrane, along with its anchoring points or surfaces (Figure 1i). An additional and thicker (3 μm) layer of electroplated gold constitutes the eighth and last layer, that is present on the RF signal and ground surfaces enclosing the MEMS device, as well as on parts of the suspended membrane intended to be more rigid. In a subsequent phase, the sacrificial layer will be removed through oxygen plasma etching, leading to the release of the RF-MEMS bridge.

The fabrication process described above is slightly more intricate when compared to procedures outlined in recent studies [40,41]. In fact, in both cases, gold is employed as structural material, but while in [40] the only other conductive layer is made of titanium–tungsten, no other conductive layer is adopted in [41]. On the one hand, such fabrication processes involve a reduced complexity, but also intrinsic limitations. Indeed, our fabrication process comes with substantial benefits, including an expanded range of design possibilities during the planning stage (different conductive layers for underpasses or overpasses) and the ability to maintain effective physical separation between the paths of the DC signal and the routed RF signals (multiple oxide layers).

## 3. The Manufacturing of the Membranes for RF-MEMS Switches and Their Mechanical Response

The adopted class of membranes relies on meandered support beams to achieve a reduced spring constant along the vertical direction for the movable structure, and thus a reduced actuation voltage (or pull-in). For this purpose, the key design feature of the proposed structure is represented by its meandered beams.

Commencing with the layout depicted in Figure 2a, various crucial layers can be discerned, with the initial layer of the fabrication process being the polycrystalline silicon. This layer constitutes both the central buried electrodes and the lateral decoupling resistors (in red). The central buried electrodes, positioned beneath the membrane’s perforated sections, are designed to induce the membrane’s collapse. Meanwhile, the lateral decoupling resistors prevent the diversion of RF signals toward the ground planes when the membrane is activated.

The multi-metal layer is utilized for implementing both the central and interrupted RF signal lines (in blue), featuring terminals enclosed by rectangular contact pads made of evaporated gold. These regions are visible in Figure 2a,d and serve as the contact surfaces with the collapsed membrane, constructed using two layers of electroplated gold. The first layer forms the primary structure of the central membrane, while the second, thicker layer is positioned specifically over surfaces intended to provide greater stiffness, such as anchors and borders of the movable membrane.

As visible in the following figures, the class of membranes is characterized by beams with uniform meanders, except for the terminal one, which was elongated to achieve a progressively lower pull-in voltage. Three devices (Dev 1, Dev 2, and Dev 3) were studied with the modifications described in Figure 2. In particular, Figure 2a reports the overall footprint of the membrane (780 × 270 μm^2^), consisting of a central rectangular plate measuring 380 × 170 μm^2^ and beams formed by uniformly meandering segments measuring 50 μm by 25 μm (as indicated by A and B in Figure 2b) with a width of 10 μm. Such dimensions are the result of the desire to balance the elongation introduced by the meandered beams with the size of the central plate, aiming at a movable structure characterized by a moderate footprint. The second device, partially shown in Figure 2c, features non-uniform meandered beams, achieved by doubling the length of each terminal meander. In the third device, depicted by the beam in Figure 2d, the length of the terminal meander was increased to four times its original length, and the lateral dimension of the meanders (F in Figure 2d) was slightly enlarged to maintain a minimum distance of 10 μm between the contact pads and the beams. As a result, the target pull-in voltage of the three membranes was progressively lowered without a substantial increase of the footprint. All the parameters reported in Figure 2 are detailed in Table 1, while additional considerations concerning the discussed structures can be found in [42].

The devices were simulated by the finite elements method (FEM) in the Ansys Workbench 2021R2 software environment, while the measurements were conducted by a manual probe station and two digital multimeters. The initial multimeter supplied the DC bias voltage and established the ground reference for the device under test (DUT) pads. Simultaneously, the second multimeter maintained a restricted voltage drop between the input and output terminals of the central RF signal line, which is interrupted when the structure is in its rest position. This was conducted to assess the resistance of the line and the level of the DC current flowing. Specifically, when the DC bias attains the pull-in voltage, the membrane’s collapse establishes a conductive path along the RF line and induces a sudden decrease in resistance (accompanied by an increase in DC current) along the RF line.

Since the DC bias was applied by increments of 1 V, the values presented in Figure 3 demonstrate a favorable agreement between the simulated and measured outcomes. Notably, while the measured pull-in voltages for Dev3 and Dev1 (9 V and 7 V) are slightly higher than the simulated values (8 V and 5 V, respectively), the measured actuation of Dev2 aligns with the anticipated performance.

The slight discrepancy observed in the Dev3 and Dev1 switches could be attributed to the heightened stiffness of their movable membranes, resulting from residual tensile stress introduced during the fabrication process. In fact, the stacking of different layers and the temperatures involved in their deposition cause a built-in stress within the layers. Compressive or tensile, such stress manifests itself through the buckling or the tensioning of the membrane, respectively. In both cases, a variation in the observable pull-in voltage is the inevitable consequence, with an increase of it being the consequence of the latter. A tensile stress characterized the fabricated samples since the measurements indicated initially pull-in voltages that exceeded those shown in Figure 3b by 10–15 V. Subsequently, a re-baking thermal treatment was applied, involving the devices being maintained at 200 °C for a few hours and then slowly cooled, resulting in the release of most of the initial residual stress. The reported measurements were conducted thereafter. Additional considerations concerning the electromagnetic behavior of the considered membranes can be found in [42].

## 4. Series and Shunt Attenuation Cells

The discussed set of membranes was adopted to develop the layout of different low-voltage attenuation cells. Concerning the topology of these cells, both series and shunt resistors were considered, following the approach reported in [30]. In such a work, single attenuation cells are presented, each featuring a series or a shunt resistor loading the RF signal line. As reported in the schematics of Figure 4, the resistors either load the line or are short-circuited based on the movement of clamped–clamped membranes. Depending on the architecture of the cell, the actuation of the membrane may short-circuit the series resistor or load the RF signal line with resistive shunt paths. Starting from the layout, 3D models of the two variants of cells were developed and the width, length, and sheet resistance of the resistors were parametrized in order to optimize them.

### 4.1. Test Bench Cells

As starting point, the design of the abovementioned attenuation cells was optimized to determine a specific value of attenuation (−2 dB, −3 dB, and −5 dB) at the center frequency (25.87 GHz) of the 24.25–27.5 GHz band, which was chosen as the practical use case since it represents the higher frequency range allocated to 5G communications in Europe (N258 band). As for the membranes reported in [30], the membranes also employed in the cells displayed in Figures 5, 7 and 8 are characterized by a substantial actuation voltage (~45 V), since this kind of membrane was used as the initial test bench. It should be remembered that the attenuation imposed on the travelling RF signal depends practically only on the dimensions and material properties of the resistors, while the movable structure is meant to solely short-circuit the series resistor or activate shunt resistors upon its collapse.

Among the most salient features of the layout reported in Figure 5a, it is possible to distinguish the central and buried poly-silicon resistor (in red) along the RF signal line, which extends along the multi-metal buried sections (in blue) and the sections of electroplated gold (in dark green). The operation of the device involves a default attenuation state due to the series resistor along the RF line, which can be short-circuited by actuating the membrane. As visible in Figure 5b, in the default attenuation state this cell introduces an attenuation level of −2 dB thanks to its 63 × 113 µm^2^ (length × width) resistor and a sheet resistance of 100 Ohm/Sq of the poly-silicon composing the resistor. The return loss and insertion loss curves are quite flat, and the range of their values is satisfying along the entire 30 GHz range (Figure 5a,b).

Unfortunately, in order to fabricate this and other attenuation cells, discussed in the following at the first available opportunity, they were added to the wafer layout containing devices to be fabricated as part of another project. A proper doping of the poly-silicon layer was not important for the devices developed during that project, and delays during the fabrication process caused the situation in which the poly-silicon layer of the devices contained in such wafer layouts underwent a coarse doping just to lower the basic sheet resistance of the basic poly-silicon. As a result, the sheet resistance of some of the following devices turned out to be in the 600–800 Ohm/Sq range, depending on the particular wafer, instead of the intended 100 Ohm/Sq, with a consequent alteration of the achieved performances.

As visible in Figure 6, the measurement conducted along the 5–40 GHz interval in both the OFF (resting membrane, attenuation) state and ON (actuated membrane) state displayed insertion loss values that are far from the ones of the optimized model of Figure 5. In particular, the measured insertion loss in the OFF state (−7.6 dB) and the related excessive return loss curve (−5.22 dB) at 25.87 GHz are due to a sheet resistance of 800 Ohm/Sq of the poly-silicon composing the 63 × 113 µm^2^ resistor. In fact, as visible in Figure 6a, the attenuation cell, simulated by taking into account a sheet resistance of 800 Ohm/Sq, exhibits return loss and insertion loss curves that approximated the measured curves with a high degree of agreement. This proves that the measured performance is due to a different sheet resistance value characterizing the fabricated samples, while a doping of the poly-silicon layer in compliance with the intended value would have determined the desired performances. In addition to this issue that affected the device behavior primarily in its attenuation state, it is possible to notice in Figure 6b that the measured insertion loss (−0.69 dB) is better than the simulated one, even in the case of intended simulated behavior (−0.77 dB). On the other hand, the measured return loss is slightly worse (−14.07 dB) than the simulated ones, in both the cases of 100 and 800 Ohm/Sq, −17.75 dB and −18.47 dB, respectively.

From a general viewpoint, it is worth noting that fabrication non-ideality concerning the improper doping of the polysilicon layer does not affect the overall reliability and the lifetime of the device, which depends solely on the structural integrity of the movable membrane. As visible in Figure 6a, the improper doping only causes a different resistivity, and this affects the performance of the sample when the RF signal travels along the resistor, determining a different attenuation level (and return loss) as compared to the intended one. When the resistor is short-circuited upon actuation of the membrane, it causes almost no attenuation to the signal amplitude, so that an improper doping implies minimal variations to the expected performances of the device in this operational state. This is demonstrated by the minimal discrepancy between the measured (−0.69 dB) and the simulated (−0.80 dB) values of insertion loss displayed in Figure 6b.

A cell featuring shunt resistors was adopted in order to achieve a −3 dB attenuation level at the center frequency of the desired frequency interval. In this topology, the signal experiences no attenuation when the membrane is in the rest position (OFF state), while part of the signal power is dissipated along the shunt resistors connecting the membrane to the ground planes when the membrane is actuated (ON state). As a result, the desired attenuation level can be achieved by controlling the dimensions and the sheet resistance of the poly-silicon resistors. In the case of the cell depicted in Figure 7a, an attenuation level of −3 dB can be reached by 93 × 28 µm^2^ (length × width) resistors and a sheet resistance of 100 Ohm/Sq. Concerning the simulated scattering curves of Figure 7b,c, it is worth noting the flatness of the return loss curve along the entire frequency interval and the precise attenuation level reached at the center frequency in the ON state. On the other hand, the return loss and insertion loss curves in the OFF state could be improved by further optimization of the CPW structure.

Another optimized version was developed to verify the maximum attenuation level that could be achieved by means of this shunt topology used as test bench. The resulting layout is depicted in Figure 8a, whose main feature is represented by the 39 × 74 µm^2^ (length × width) shunt resistors, sized on the basis of a 100 Ohm/Sq sheet resistance. As visible in Figure 8c, a maximum attenuation level of almost −5 dB can be achieved by this topology, without substantial degradation of the return loss curve, which lies between −9.94 and −10.14 dB within the 24.25–27.5 GHz interval. Also in this case, the return loss and insertion loss curves of the OFF state could be improved by further optimization of the CPW parameters, by varying the width of the RF signal line and the lateral gaps and by the introduction of a dedicated circuital section for the compensation of the reactive impedance.

Fabricated samples of both the −3 dB and the −5 dB cell variants were characterized, and the post-measurement simulations showed that the sheet resistance of the fabricated −3 dB and −5 dB cell variants also corresponds to 800 Ohm/Sq. Thus, only the simulated S parameters of the −3 and −5 dB variants were reported for sake of brevity.

### 4.2. Low Pull-In Cells

For each of the abovementioned attenuation levels, variants equipped with the 5 V, 7 V, and 9 V membranes of Figure 3 were fabricated and characterized. The layout and the operation of the cell displayed in Figure 9a does not substantially differ from the one displayed in Figure 5, except for being a “dielectric-less” design. In fact, in this design, the layers of silicon oxide covering the buried electrodes in correspondence of the squared areas of the movable membrane are meant to be removed to avoid the phenomenon of charge accumulation/trapping. In their place, quadrangular stopping pillars of poly-silicon, covered by the multi-metal and evaporated-gold layers were placed, to avoid the contact between the fixed electrodes and the movable membrane. The present attenuation cell was optimized by maintaining the same dimensions of the series resistor (63 × 113 µm^2^) and varying the sheet resistance of the poly-silicon. In particular, by a 130 Ohm/Sq sheet resistance this design could achieve an attenuation level of nearly—2 dB (Figure 9b), with rather flat curves of return loss and insertion loss along the whole considered interval. When the membrane is actuated and the resistor is short-circuited in its ON state (Figure 9c), this variant shows a quite limited insertion loss and a return loss ranging between −11.46 and −10.15 dB along the 24.25–27.5 GHz.

As for the previous cells, the poly-silicon of the cells featuring the low pull-in membranes was also subject to coarse doping. In particular, the poly-silicon, on the wafer containing the fabricated samples that were characterized, is marked by a sheet resistance of 600 Ohm/Sq, instead of the planned 130 Ohm/Sq. As for the abovementioned −2 dB variant, the measured performances inevitably differ from the planned ones and the outcomes of the model simulated by taking into account a 600 Ohm/Sq sheet resistance are in good agreement with the measurements. In particular, the simulated curves of the OFF state (Figure 10a) qualitatively match the measurements up to 31 GHz, after which the measurements become noisier, and growing ripples appear. Unlike the previous −2 dB variant, when the series resistor is short-circuited in the ON state, the measured return loss is generally better than the simulated ones (both the intended 100 Ohm/Sq, −10.79 dB, and the unplanned 600 Ohm/Sq, −10.80 dB) along the major part of the frequency interval. On the other hand, the measured insertion loss (−1.05 dB) is worse than the simulated ones.

Concerning the variants aiming at a −3 dB attenuation level, a cell equipped with the membrane with a 7 V pull-in voltage is reported in Figure 11a. Also in this case, during the design phase the dimension of the shunt resistors was kept the same as the ones of the −3 dB test bench model and the sheet resistance was varied to reach an attenuation of −3 dB at the desired frequency. As a result, 93 × 28 µm^2^ (length × width) shunt resistors marked by 80 Ohm/Sq sheet resistance characterize this cell. As with the previous cell, when no attenuation is imposed on the signal (OFF state), the measured return loss is definitely better than the simulated one, while the measured insertion loss is generally greater than the simulated one (~0.7 dB), and a good qualitative agreement between the two curves is noticed in Figure 11b. The same trend marks the curves displayed in Figure 11c: a discrepancy of ~0.7 dB can be seen among the attenuations, while the return loss of the fabricated sample is definitely better than the simulated one along the entire frequency interval.

Concerning such discrepancies, as a general remark, it is worth noting that in the case of electromagnetic full-wave simulations of devices meant for high frequencies, some simplifications are necessary to introduce to the 3D model to ease the meshing process and the convergence of solution. Starting from the emulation of the fabrication process, it is not always possible to replicate faithfully some operations (e.g., isotropic deposition), so that slight modifications to the thickness of some layers is performed to achieve overall correct stacking. For the same reason, the deformation of the membrane due to the sacrificial layer cannot be easily reproduced, therefore flat membranes are adopted in the 3D model and moved to model the actuation and its rest position, which slightly alter the amount of capacitive coupling between the membrane and signal line during both states. In addition, bodies which are not strictly necessary for the functioning of the device (in its operations in the AC regime), such as the DC decoupling resistors, are suppressed to ease the meshing process and the convergence of the solution. All these simplifications can be cast as a set of distributed complex impedances that globally play a role as I/O mismatches (referring to the return loss) and/or adjunctive losses (referring to the insertion loss). These effects, combined with possible non-idealities in the multi-step manufacturing process, motivate the differences between simulation and measurements, and can be investigated and mitigated in further steps of optimization.

The abovementioned cells featuring low pull-in membranes display an interesting behavior, for which the return losses of the fabricated samples are significantly limited as compared to the simulation outcomes, despite a minimal discrepancy between the simulated and the measured insertion loss curves. However, the substantial agreement between the displayed performances of the different cells proves that the expected performance can be achieved for the considered designs, providing a faithful fabrication process. As visible in Table 2, the proposed devices combine a small footprint with a reduced actuation voltage. In addition to the analog implementations, which belong to another operating principle, the proposed devices represent the first attempt to introduce low pull-in membranes in the field of RF-MEMS attenuators. Moreover, it is worth noting their reduced footprint as compared to the other 1-bit implementations. Due to their footprint, such attenuator cells could be effectively combined to form multi-bit architectures with smaller areas as compared to the listed counterparts.

From a critical perspective, the simulations and the measurement outcomes showed that by such basic design, maximum attenuation levels of about −3 dB and −5 dB can be achieved, by series and shunt design, respectively. For larger amounts, the other topologies and combinations described in the second part of the present paper should be considered.

## 5. Conclusions

In this paper, different variants of basic and compact attenuator modules for beamforming applications are described and critically assessed. Starting from the 1-bit modules targeting attenuation values of −2, −3, and −5 dB, they are presented in different variants, featuring movable membranes characterized by substantial actuation voltage (~45 V) or low actuation (5–9 V). More specifically, the cells equipped with the membranes characterized by the more substantial actuation voltage were adopted as the initial test bench, in order to determine the attenuation levels achievable by a series or two shunt resistors following their optimization in terms of dimensions and sheet resistance. Afterwards, cells equipped with membranes characterized by low actuation voltages and based on the test bench models were designed and optimized in their sheet resistance, after which variants of the different topologies were fabricated and assessed.

Beyond the fabrication non-idealities that affected the discussed samples, a substantial agreement marked the simulated and the measured performances of the attenuation cells up to 40 GHz. Concerning the observed non-idealities, the post-measurement simulations highlighted the difference between the intended and the actual sheet resistance marking the fabricated samples, due to accidental factors. The advantages of the proposed modules consist in a substantial compactness and, depending on the membrane, in a reduced driving voltage. On the other hand, a faithful replication of the intended value of sheet resistance in the fabricated samples proved to be essential to achieve the desired performances. Under the premise of such a faithful fabrication process, further improvements to the modules in their present design could be the optimization of the CPW transmission line as well as the addition of circuital sections devoted to the compensation of the reactive impedance represented by the overall module. Concerning the practical limitations of the considered modules, the behavior of the proposed cells based on series and shunt resistors proved that they could be adopted to achieve fine attenuation levels up to nearly −3 dB and −5 dB, respectively. Higher attenuation levels could be reached only by combining the series and shunt resistors in higher-order topologies, such as T-shaped or Pi-shaped resistors. However, the overall compactness of the considered designs suggests that they could be effectively employed as basic building blocks for more complex and low-voltage networks, aiming at a wider attenuation range.

## Figures and Tables

**Figure 1 sensors-24-02308-f001:**
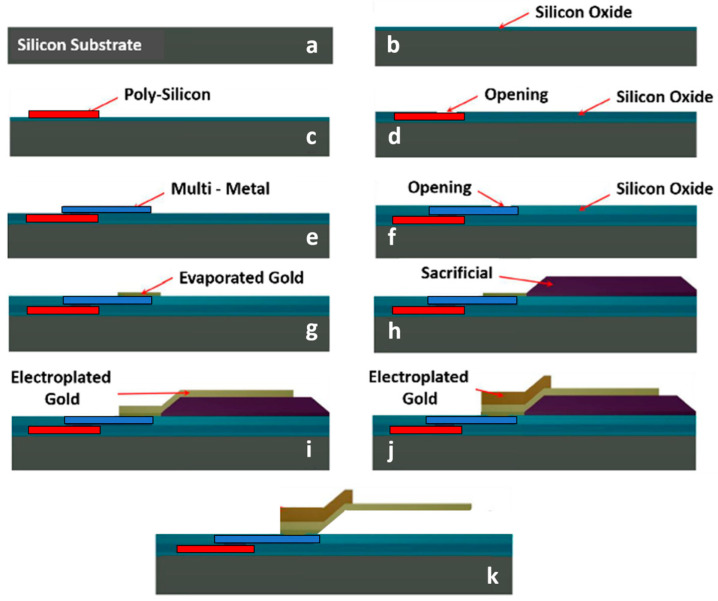
Surface micromachining process used for the fabrication of the reported devices. In particular, it is worth noticing (**a**) the basic substrate, (**b**) the basic oxide covering the wafer, (**c**) the poly-silicon layer, (**d**) the subsequent openings, (**e**) the multi-metal layer, (**f**) the subsequent openings, (**g**) the evaporated gold, (**h**) the sacrificial layer, and the (**i**,**j**) two layers of electroplated gold that form the movable structure.

**Figure 2 sensors-24-02308-f002:**
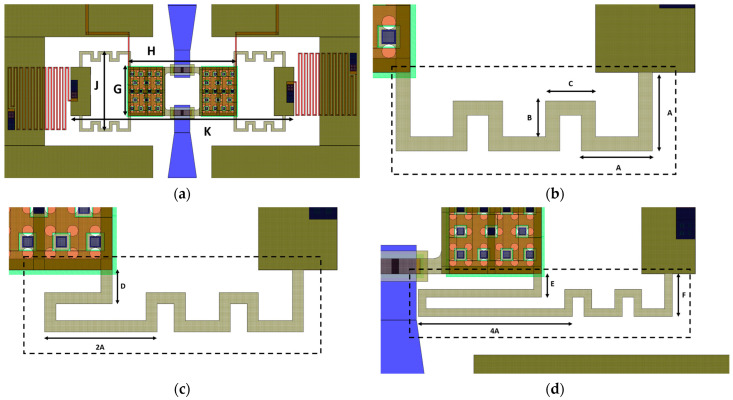
Main design parameters of the employed class of membranes, including (**a**) overall view, and detail of the beams (evidenced by dashed rectangle) characterizing the membranes of (**b**) Dev1, (**c**) Dev2, and (**d**) Dev3, whose support beams are characterized by progressively elongated meanders to achieve an increasingly reduced actuation voltage.

**Figure 3 sensors-24-02308-f003:**
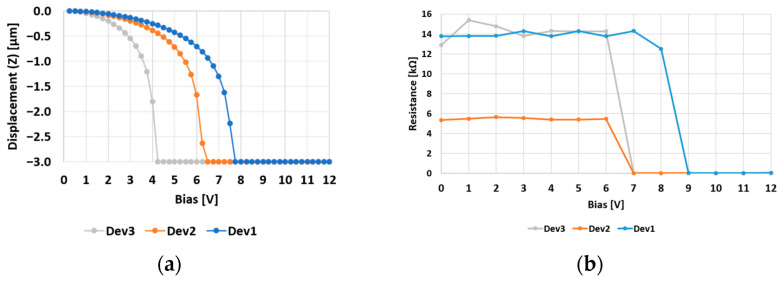
(**a**) Simulated vertical displacement and (**b**) measured pull-in voltage of the three membranes considered in Figure 2.

**Figure 4 sensors-24-02308-f004:**
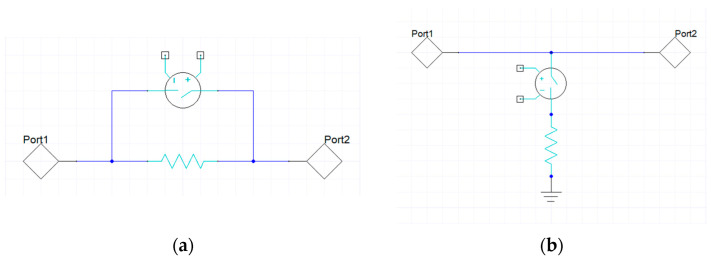
Equivalent topology of the following (**a**) series and (**b**) shunt attenuation cells.

**Figure 5 sensors-24-02308-f005:**
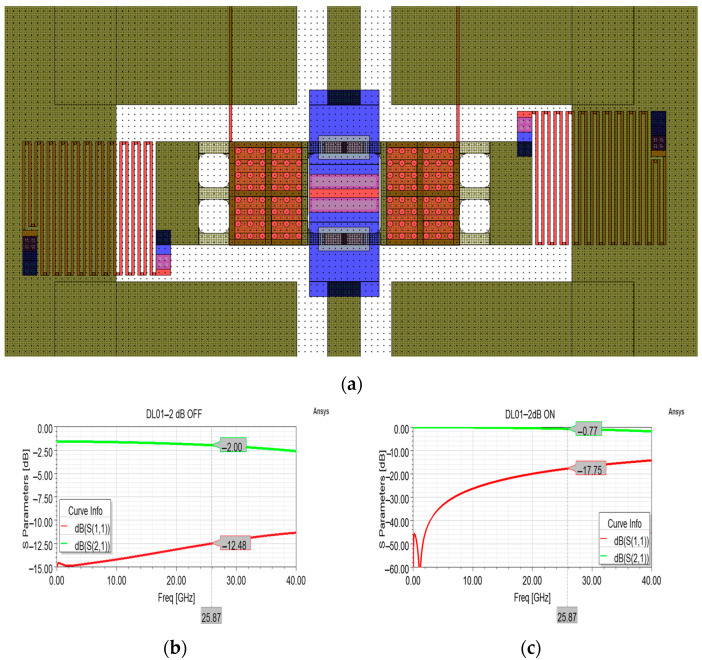
(**a**) Overall layout of the attenuation cell featuring a single series resistor for a desired attenuation level of −2 dB; (**b**) simulated S parameter curves in the OFF state, and (**c**) ON state. In particular, it is worth noting the central resistor (in red), that can be short-circuited by the actuation of the membrane.

**Figure 6 sensors-24-02308-f006:**
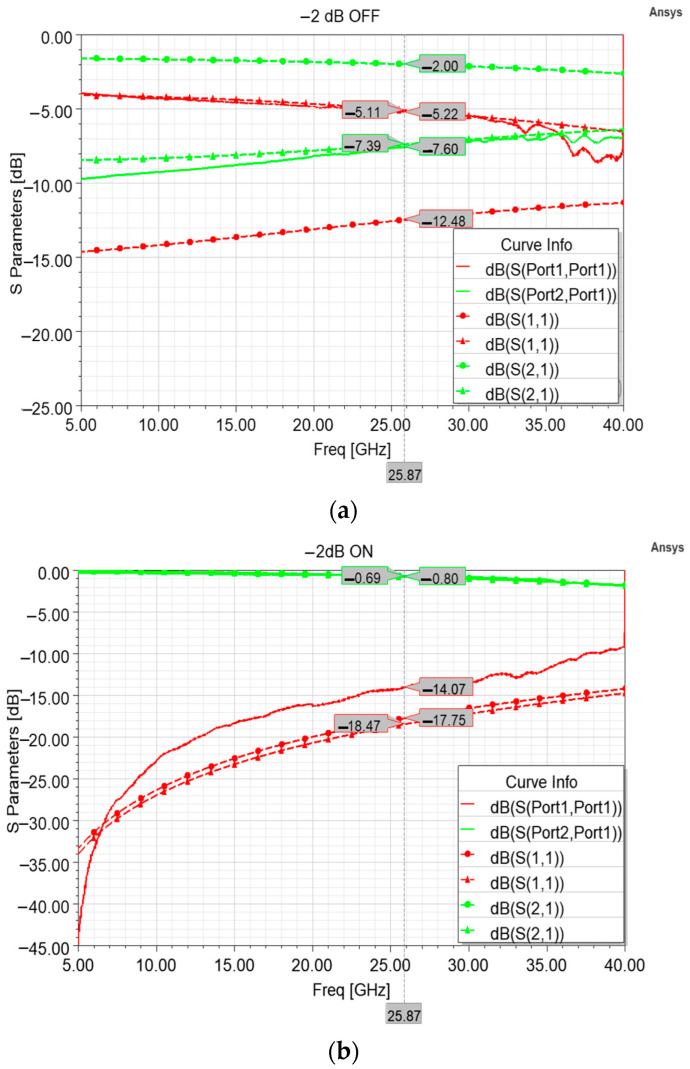
Comparison between different insertion loss (green) and return loss (red) curves. In particular, measured curves (solid lines) are compared to the simulated curves (dashed lines) taking into account the proper 100 Ohm/Sq sheet resistance (circle marks), and the simulated curves taking into account the actual 800 Ohm/Sq sheet resistance of the fabricated devices (triangle marks), in both the default attenuation state (**a**) and non-attenuation state (**b**).

**Figure 7 sensors-24-02308-f007:**
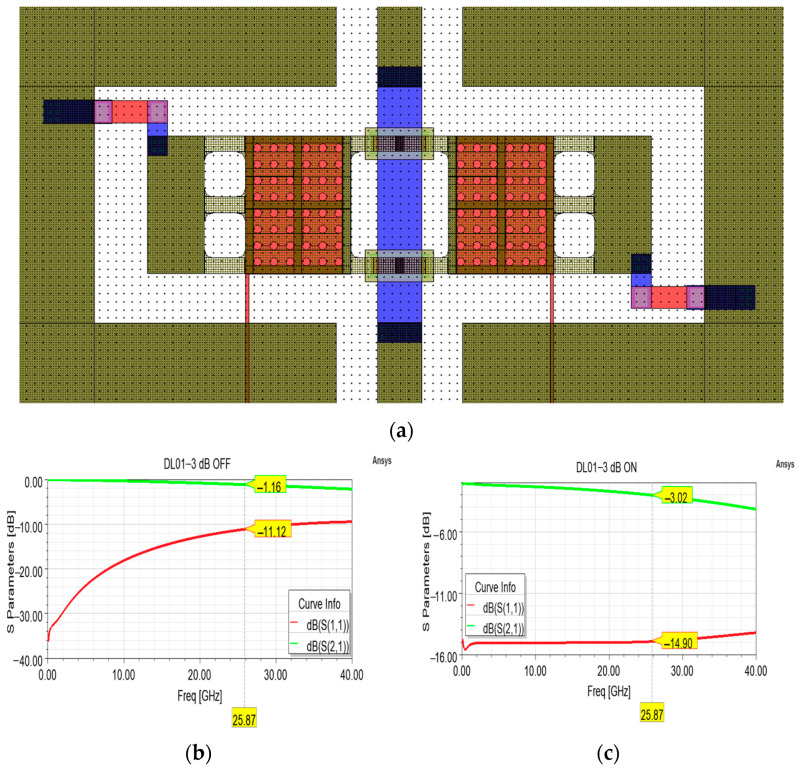
(**a**) Overall layout of the attenuation cell featuring shunt resistors for a desired attenuation level of −3 dB; (**b**) simulated S parameter curves in the OFF state, and (**c**) ON state.

**Figure 8 sensors-24-02308-f008:**
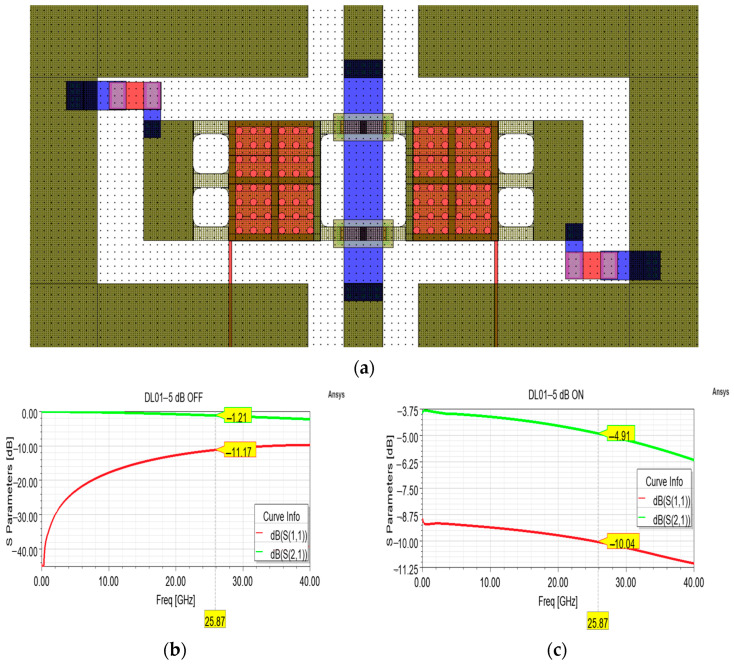
(**a**) Overall layout of the attenuation cell featuring shunt resistors for a desired attenuation level of −5 dB; (**b**) simulated S parameter curves in the OFF state, and (**c**) ON state.

**Figure 9 sensors-24-02308-f009:**
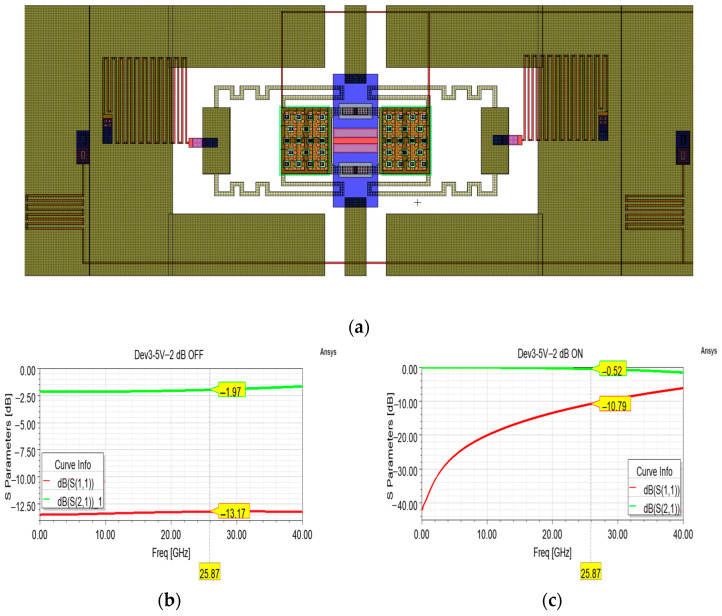
(**a**) Overall layout of the attenuation cell featuring a single series resistor for a desired attenuation level of −2 dB; (**b**) simulated S parameter curves in the OFF state, and (**c**) ON state.

**Figure 10 sensors-24-02308-f010:**
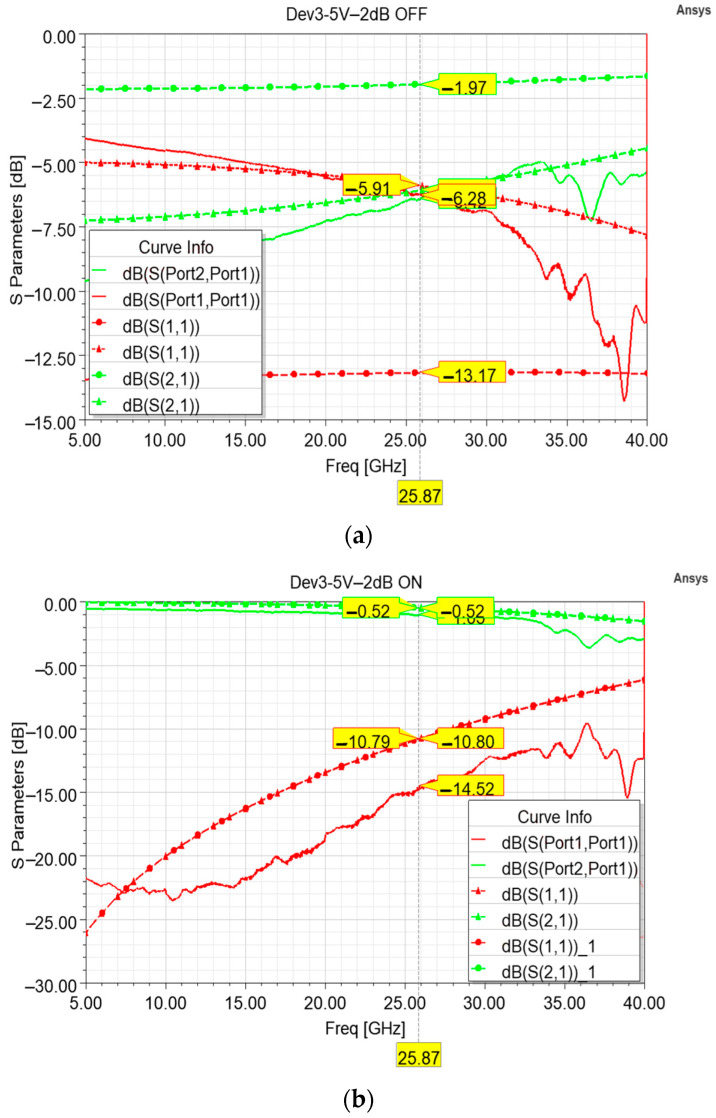
Comparison between different insertion loss (green) and return loss (red) curves. In particular, the measured curves (solid lines) are compared to the simulated curves (dashed lines) taking into account the planned 130 Ohm/Sq sheet resistance (circle marks) and the simulated curves taking into account the actual 600 Ohm/Sq sheet resistance of the fabricated devices (triangle marks) in both the default attenuation state (**a**) and non-attenuation state (**b**).

**Figure 11 sensors-24-02308-f011:**
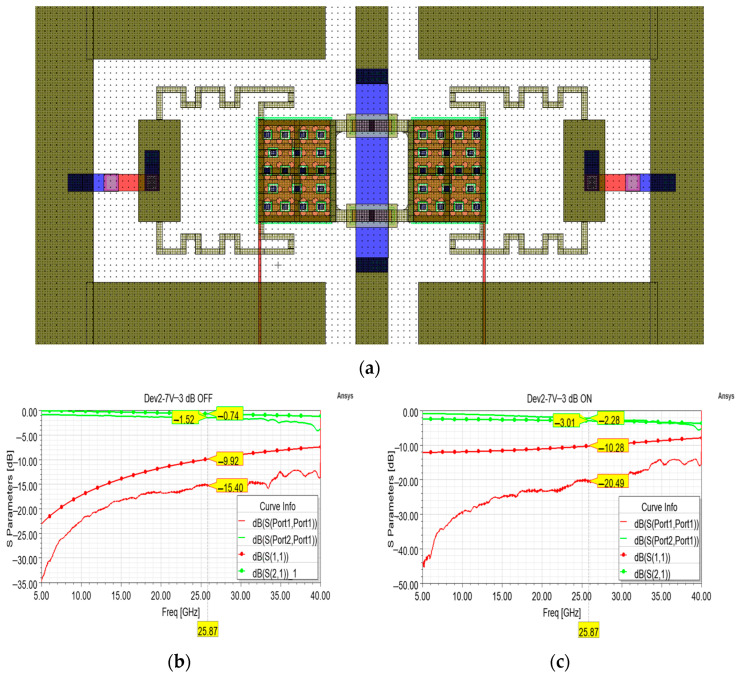
(**a**) Overall layout of the attenuation cell featuring shunt resistors for a desired attenuation level of −3 dB; (**b**) comparison between simulated (circle marks) and measured S parameter curves (no marks) in the OFF state, and (**c**) ON state.

**Table 1 sensors-24-02308-t001:** Design parameters of the represented devices.

Parameter	Value [µm]	Parameter	Value [µm]
A	50	B	25
C	35	D	25
E	30	F	55
G	170	H	380
J	270	K	780

**Table 2 sensors-24-02308-t002:** Quantitative comparison between the features of the most significant RF-MEMS attenuators populating the literature of the last decade.

Work	Control	Size [mm^2^]	Driving Voltage [V]	Operational Frequency [GHz]	Achievable Attenuation [dB]	Tuning Step/ Resolution [dB]
[2]	analog	2 × 5	0.5	22–27	1–16	0.2
[3]	analog	3.8 × 3.1	9.8	58–62	10–25	-
[30]	1-bit	0.7 × 2	48–50 48–50	0–110 0–110	5–9 (series) 3.5–11.5 (shunt)	switched
[31]	8-bit	1.95 × 3	45–50	0–80	10–45	-
[33]	3-bit	2.45 × 4.34	20–30	0–20	10–70	10
[34]	3-bit	3.2	60	0–20	0–35	5
[35]	4-bit	2.15 × 7.5	51	0–20	1.65–17.02	-
[43]	1-bit	2.1 × 4.4	40	0–20	11.3	switched
This Work	1-bit	0.69 × 1.35	7, 9, 45	24.25–27.5	2 (series) 3, 4.9 (shunt)	switched

## Data Availability

Data are contained within the article.

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
