# Peer review of "Discussion and Demonstration of RF-MEMS Attenuators Design Concepts and Modules for Advanced Beamforming in the Beyond-5G and 6G Scenario—Part 1"

_sensors, 2024, doi:10.3390/s24072308_

Round 1
Reviewer 1 Report
Comments and Suggestions for Authors
The manuscript is prepared in good way. Below are some comments:
In Figure 1, please mark different colors for different materials, add the final image after the sacrifical layer is released.
Describe the comparison between the present process and the process in the paper. [Line: 173-176]
Identify the key components in Figure 2.
In Figure 6, mark the simulation curve and measurement curve.
How to optimize the CPW structure? [Line: 328-330]
In Figure 10 (b), there are 6 types of curve information, but only 4 curves in the graph?
In Figure 11, are the measured and simulated curves for the S-parameter? Please mark clearly.
Is 25.87GHz the central frequency for your design? Please explain the reason for this.
Comments on the Quality of English Language
Minor editing of English language required.
Author Response
First, we hereby thank Reviewer 1 for the positive feedback returned to us. We addressed all the points in the revised version of the paper (highlighted in YELLOW) and below we provide point-by-point replies to them.
In Figure 1, please mark different colors for different materials, add the final image after the sacrifical layer is released.
==>> Different colors have been assigned to represent different materials, and the final image after the sacrificial layer release has been included. We have redesigned the Polysilicon and the Multi-Metal layers, which can be distinguished more easily now within the stack.
Describe the comparison between the present process and the process in the paper. [Line: 173-176]
==>> The comparison between our process and the mentioned processes has been expanded within the 174-181 lines, detailing the advantages characterizing our process.
Identify the key components in Figure 2.
==>> We added a remark within the text [lines 188, 190-191] to specify that the key components of the described movable structures are the meandered support beams. Additional mentions have been added in lines 192, 195, 200, 201 to let the reader identify more easily the parts of the device contributing to its proper operation.
In Figure 6, mark the simulation curve and measurement curve.
==>> The visual difference between the measurement and the simulation curves has been increased by using dashed lines and marks (circular and triangular) for the simulation curves and solid lines (no marks) for the measurement curves. The difference has been highlighted in the caption [lines 337-341].
How to optimize the CPW structure? [Line: 328-330]
==>> In line 363-365 we added a clarification to specify that the optimization of the CPW structure would focus on its dimensional parameters and on the insertion of a dedicated circuital section.
In Figure 10 (b), there are 6 types of curve information, but only 4 curves in the graph?
==>> We applied the same modifications as for Fig. 6, by using solid lines for the measurement curves, and dashed lines (circular and triangular marks) for the simulation curves. By these modifications, it is now possible to notice that the simulation curves in case of intended and obtained doping show a minimal difference in the S-Parameters of the module when the resistor is short-circuited.
In Figure 11, are the measured and simulated curves for the S-parameter? Please mark clearly.
==>> We added a clarification in the caption [lines 444-445] to clearly mark the difference between the displayed curves.
Is 25.87GHz the central frequency for your design? Please explain the reason for this.
==>> The reason for the choice of that frequency band has been expanded in lines 274-277.
Reviewer 2 Report
Comments and Suggestions for Authors
Authors present an interesting topic of RF-MEMS attenuator design at 24-28 GHz range. The writing is sketchy and technical details are largely missing so that it is very difficult to understand the manuscript except for those who are studying or have studied similar structures. Furthermore, the organization of the manuscript is not logical for a research paper.
This reviewer's opinion is that the manuscript needs to be rewritten and resubmitted.
(1) With the organization of the submitted manuscript, it is confusing, or even difficult to understand the concept and structure of the proposed device.
Please revise the organization of the manuscript for improved readability. Possible organization is as follows.
1. Introduction
2. Device Structure and Operating Principles: Device structures may be in a block diagram which include resistors and switches. How to control attenuation.
3. Device Layout and Dimensions: Denote attenuator components such as resistors, switches, lines.
4. Fabrication Processes
5. Simulation Results and Discussions
6. Conclusions
(2) '2. Fabrication': In Figure 1, the meaning of 1° and 2° is not clear. Is it a standard technical notation?
(3) '3. Membranes'
- The section title does not represent the section contents. Is the section about the proposed MEMS switch?
- The MEMS switch structure is described very coarsely.
- Figure 3 needs to be correlated with the structure of Figure 2 for easy understanding by readers. The description is stekchy and details are missing. In the worst case, Figure 3 can be meaningless.
(4) '4. Series and Shunt Attenuation Cells'
- In Figure 5, denote the attenuation cells and switches.
- In Figure 6, it no clear how to control attenuation with the layout of Figure 2.
- The structures in Figure 5(a), 7(a), 9(a), and 11(a) are different from the structure in Figure 2(a). What is Figure 2(a) for then?
- Figure 6(a): The control of four levels of attenuation is not clear without an overall block diagram.
- Please comment on the variation of attenuation with frequency which is very large in some cases (e.g., -11.12 dB curve in Figure 7(b); -10.79 dB curve in Figure 9(c)).
(5) '7. Conclusions': Conclusions seem to be too similar to 'Abstract'. Please rewrite the wording for summary and include some more writing on the findings (maybe containing advantages, disadvantages, and limitations of the proposed device) and possible topic for additional research.
Author Response
We hereby thank Reviewer 2 for the feedback and comments provided. We marked the changes applied to the revised article highlighting the corresponding text in RED. We did our best to address the remarks to the widest extent, despite the boundaries imposed by a scientific article (rather than a longer manuscript) making it not possible to step into excessively basic technical details.
This reviewer's opinion is that the manuscript needs to be rewritten and resubmitted.
(1) With the organization of the submitted manuscript, it is confusing, or even difficult to understand the concept and structure of the proposed device.
Please revise the organization of the manuscript for improved readability. Possible organization is as follows.
- Introduction
- Device Structure and Operating Principles: Device structures may be in a block diagram which include resistors and switches. How to control attenuation.
- Device Layout and Dimensions: Denote attenuator components such as resistors, switches, lines.
- Fabrication Processes
- Simulation Results and Discussions
- Conclusions
We thank Reviewer 2 for suggesting an alternative arrangement of the paper. We could not address this point to the full, as we believe that an organization of the work in which the technology part is first explained is more logical and easier to understand for the reading audience.
(2) '2. Fabrication': In Figure 1, the meaning of 1° and 2° is not clear. Is it a standard technical notation?
==>> In Figure 1 we deleted the notation “1°” and “2°” since it could mislead the reader. As reported in the text describing the fabrication process, the notation referred to the first and second layer of electroplated Gold. After this modification, the picture should be clearer. In addition, the different steps described in the figure have been mentioned within the text [lines 151, 157, 167] and the caption [183-185], to let the reader link more easily the described layers with their graphical representation.
(3) '3. Membranes'
- The section title does not represent the section contents. Is the section about the proposed MEMS switch?
- The MEMS switch structure is described very coarsely.
- Figure 3 needs to be correlated with the structure of Figure 2 for easy understanding by readers. The description is stekchy and details are missing. In the worst case, Figure 3 can be meaningless.
==>> We modified the title of the section to clearly specify that this section details the movable structure and its mechanical behavior, when it is employed as switch.
We displayed the layout, simulations and measurements of the movable structures in a switch configuration since it is the most common configuration in which a movable structure can be employed and found in the RF-MEMS literature. Within the text, we refer to the movable structure as to “membrane”, since “membrane” or “bridge” are common terms used to refer to clamped-clamped structures in the RF-MEMS literature and scientific community.
For sake of brevity, we highlighted only the most crucial layers and parts of the movable structure: the Polysilicon layer (for resistors), the Multi-Metal layer (for the RF signal line), and the meandered support beams, that are crucial to achieve the described low actuation voltage. For this purpose, we stressed in lines [190-192, 226-227] the importance of these layers and parts, while references to Fig. 2 were added in lines 195, 200, 201. Still for brevity, we omitted additional considerations about the described membranes not strictly related to their functioning (in the context of the subsequently described attenuation modules).
Following your suggestion, we added a specification in the caption of Fig. 3 to make it clear that the displayed results are related to the membranes of Fig. 2. In addition, we added a couple of comments regarding the functioning and the measurement of the displayed structures in lines 235-238.
(4) '4. Series and Shunt Attenuation Cells'
- In Figure 5, denote the attenuation cells and switches.
- In Figure 6, it no clear how to control attenuation with the layout of Figure 2.
- The structures in Figure 5(a), 7(a), 9(a), and 11(a) are different from the structure in Figure 2(a). What is Figure 2(a) for then?
- Figure 6(a): The control of four levels of attenuation is not clear without an overall block diagram.
- Please comment on the variation of attenuation with frequency which is very large in some cases (e.g., -11.12 dB curve in Figure 7(b); -10.79 dB curve in Figure 9(c)).
==>> Following your suggestion, we denoted the resistor and the role of the movable membrane in the caption of Fig. 5.
In Fig. 6 we changed the appearance of the displayed curves to clarify that the measured performance (solid lines) is compared to the simulated performance (dashed lines). In particular, it is now clearer that the discrepancy between the intended performance (dashed line, circle marks) and the measured performance (solid line) is due to a different sheet-resistance value of the Polysilicon layer of the fabricated devices. When considered in the simulations, the different and considered value of sheet-resistance (800 Ohm/Sq) determines a simulated performance (dashed line, triangle marks) that fits the measured performance. This proves that the discrepancy between the intended performance and the measured performance is due to that specific variation of sheet-resistance value.
Following the comments of another reviewer, we added additional considerations about the impact of such fabrication issue on the reliability and performance of the module in lines 325-336, remarking in line 331 that the series resistor can only be short-circuited by the actuation of the membrane. Due to their intrinsic functioning, it is not possible to express the operation of the presented modules by means of a block scheme: the equivalent circuits of Fig. 4, showing the single resistors selected by the single switches, have been displayed for this purpose.
The structures of Fig. 5(a), 7(a), 8(a) do not have meandered beams, since they were used as initial test bench to develop the following attenuation modules characterized by low actuation voltage, as specified in line 279. The structures of Fig. 9(a), 11(a) are characterized by the meandered beams reported in Fig. 2 (d),(c). In Figure 2, the proposed class of membranes is displayed, showing the overall footprint (Fig. 2(a)) and the beams (Fig. 2 (b)) of the variant called Dev1. As specified in lines 373-374, modules equipped with every type of membrane (45 V, Dev1, Dev2, Dev3) have been fabricated for each considered attenuation level (-2, -3, -5 dB). Out of the 12 possible resulting fabricated variants, just a selection of them has been shown in the manuscript for sake of brevity.
About the variation of attenuation with frequency: the –11.12 curve of Figure 7 (b) is not a curve of attenuation; it is the curve showing the return loss of the module (S(1,1)) in the OFF state, namely when the membrane is elevated (rest position) and the signal travelling along the central RF signal line experiences minimal losses due to the device itself (S(2,1), -1.16 dB curve). When the membrane is actuated (ON state), the signal is attenuated (S(2,1), -3.02 dB curve) due to activation of the conductive paths featuring the optimized resistors towards the ground planes. Both the –1.16 dB and the –3.02 dB attenuation curves are quite stable along the extended 0-40 GHz interval.
Also the –10.79 curve of Figure 9 (c) is not an attenuation curve, but the return loss of the device (S(1,1)) in the ON state, when the membrane is actuated. In this state, the resistor is short-circuited by the collapsed membrane and the signal travelling along the central RF signal line experiences minimal losses due to the device itself (S(2,1), -0.52 dB curve). When the membrane is elevated (OFF state), the signal travels along the resistor along the RF line and experiences an attenuation of –1.97 dB for this reason. Both the –0.52 dB and the –1.97 dB attenuation curves are quite stable along the extended 0-40 GHz interval.
(5) '7. Conclusions': Conclusions seem to be too similar to 'Abstract'. Please rewrite the wording for summary and include some more writing on the findings (maybe containing advantages, disadvantages, and limitations of the proposed device) and possible topic for additional research.
==>> The conclusions have been modified according to your suggestion.
Reviewer 3 Report
Comments and Suggestions for Authors
This paper mentioned fabrication non-idealities affecting the sample devices but does not explore into potential mitigation strategies or the impact of these non-idealities on long-term reliability and performance.
While the paper discusses the simulation results and measurement outcomes in Figure 3, it would be beneficial to include a more detailed comparison between simulated and measured results, especially where the discrepancies occur. Discuss the reasons for these discrepancies too.
Add a comparison table that clearly outlines how the proposed work stands against the latest research.
Comments on the Quality of English LanguageMinor editing of English language required
Author Response
It is our intention to thank you for your feedback, since your input made it possible to improve our work. We addressed all the points in the revised version of the paper (highlighted in GREEN), and below we provide point-by-point replies to them.
Comments and Suggestions for Authors
This paper mentioned fabrication non-idealities affecting the sample devices but does not explore into potential mitigation strategies or the impact of these non-idealities on long-term reliability and performance.
==>> In lines 325-336 we added more extensive considerations about the impact of the described fabrication non-idealities on the reliability that affected the measured samples. We specified that the doping issue has no effect on the reliability of the device, since the latter is solely related to the lifetime of the movable structure. As specified, the obtained sheet resistance of the Polysilicon layer only determines a permanent shift in the attenuation level of the module when the resistor loads the RF signal line.
While the paper discusses the simulation results and measurement outcomes in Figure 3, it would be beneficial to include a more detailed comparison between simulated and measured results, especially where the discrepancies occur. Discuss the reasons for these discrepancies too.
==>> In lines 247-254 we detailed the dynamics related to the fabrication that determine the described discrepancy between the simulated and the measured actuation voltages, highlighting the root cause in the initial tensile stress affecting the structures, which has been mitigated by re-baking.
Add a comparison table that clearly outlines how the proposed work stands against the latest research.
==>> Table 1 has been added at the end of the last section, in which the proposed devices have been compared to recent and similar works populating the recent literature. This comparison has been briefly commented in lines 452-458, underlining the main features of the proposed devices.
Round 2
Reviewer 2 Report
Comments and Suggestions for Authors
The revised manuscript addresses all the issues raised by this reviewer's. The revised version can be accepted for publication.